# The Use of Agro-Industrial Waste Rich in Omega-3 PUFA during the Weaning Stress Improves the Gut Health of Weaned Piglets

Daniela Eliza Marin [1,*], Andrei Cristian Anghel [1], Cristina Valeria Bulgaru [1], Iulian Grosu [1], Gina Cecilia Pistol [1], Ana Elena Cismileanu [2] and Ionelia Taranu [1]

[1] Laboratory of Animal Biology, National Institute for Research and Development for Biology and Animal Nutrition, Calea Bucuresti No. 1, 077015 Balotesti, Ilfov, Romania; andrei.anghel@ibna.ro (A.C.A.); cristina.bulgaru@ibna.ro (C.V.B.); iulian.grosu@ibna.ro (I.G.); gina.pistol@ibna.ro (G.C.P.); ionelia.taranu@ibna.ro (I.T.)

[2] Laboratory of Animal Physiology, National Institute for Research and Development for Biology and Animal Nutrition, Calea Bucuresti No. 1, 077015 Balotesti, Ilfov, Romania; ana.cismileanu@ibna.ro

* Correspondence: daniela.marin@ibna.ro

**Abstract:** (1) Background: The weaning period is a very important stage in the pig life, as during weaning, the animals are very susceptible to pathogens and develop postweaning diarrhoea. The aim of our study was to counteract weaning stress and to improve piglets' gut health by using a nutritional intervention consisting of a mix of agro-industrial wastes (grapeseed, flaxseed and sea blackthorn meals) rich in omega-3 PUFA. (2) Methods: Twelve cross-bred TOPIG hybrid piglets with an average body weight of 11.25 kg were randomly distributed to one of the two experimental groups: a control group fed basic corn soybean diet (control diet) and an experimental group fed a diet with a 10% mixture of grapeseed, flaxseed and sea buckthorn meals in a ratio of 3:4:1 (GFS diet). (3) Results: the GFS diet had no effect on the performance, biochemical parameters or the total antibody synthesis. GFS diet was able to significantly reduce the concentration of proinflammatory cytokines IL-1 beta and TNF-alpha and to significantly increases the expression of junction proteins (occludin, claudin 4, claudin 7 and extracellular protein matrix) at the gene or protein level as compared with control. The presence of GFS in the diet increased the abundance of *Bifidobacterium* and *Lactobacillus* species in the colonic content as well as the concentration of propionic and butyric acids. (4) Conclusions: Taken together, our results showed that agro-industrial wastes rich in omega-3 PUFA can be used as an ecological, environmentally friendly nutritional intervention for improving the negative effects associated with the weaning stress.

**Keywords:** weaned piglets; agro-industrial waste; omega-3 PUFA

## 1. Introduction

Weaning is an important period in the pig's life associated with many nutritional, environmental or social changes as a result of the transition from milk to solid feed, separation from the sows, nonlittermate piglets mixing, transport or other environmental stress factors such as temperature or air quality changes [1]. For these reasons, the weaning crisis is associated not only with a decrease of the feed intake and weight gain after weaning but it can significantly affect the gut health of piglets with negative effects on gastrointestinal tract structure and function [2,3]. Weaning triggers an inflammatory condition of the gastrointestinal tract that leads to an alteration of intestinal epithelial barrier morphology and permeability, to a change of the microbiota composition and to the release of different effectors of the inflammatory response [4]. Weaning stress is of great economic concern for most commercial farms and the improvement of nutrition, housing or of the environmental conditions can reduce some of the negative effects of weaning [2,5]. Many

nutritional solutions have been developed in order to prevent or to reduce the disturbances associated with the weaning period [6]. Among these, many plant extracts and natural substances have been used during the weaning period for improving gut health and animal performances [7]. In the last years, the use of natural waste/by-products in animal nutrition has gained a lot of interest, taking into account the increased consumer demands for natural products associated with the necessity of the agro-industry waste reutilisation [8,9]. Agro-industry oils generate important quantities of wastes that can raise important problems for the environment if they are not properly valorised [10]. These wastes contain a high quantity of bioactive compounds, especially if the oils are extracted using green technologies [11]. They can be used as nutritional solutions for the weaning period in order to improve the digestion of nutrients, gut morphology, inflammatory response and microbiota composition and to restore intestinal homeostasis [12,13]. Polyunsaturated fatty acids (PUFA) represent important bioactive constituents of the oil industry waste [14]. Long-chain PUFA are constituents of cell membrane phospholipids [15]. PUFA act as signalling molecules and modulate the inflammatory state of different intestinal cell types, including endothelial cells, monocytes and macrophages by targeting different nuclear receptors [16]. The aim of our study was to counteract the weaning stress and to improve piglets' health by using a nutritional intervention, consisting in a mix of agro-industrial wastes (flaxseed, grapeseed and sea blackthorn meals) rich in omega-3 PUFA. We assumed that the new formula destined to weanling piglets would provide the animals a wide diversity of PUFA from different sources (flaxseed, grapeseed and sea blackthorn meals) that would ensure an easy transition of the piglets through the weaning period. The effect of the administration of omega-3 rich wastes in weaned piglets was investigated on the general animal health (body weight, feed intake, feed:gain ratio, plasma biochemistry) and on the gut health (inflammation, intestinal epithelial integrity, selected microbiota composition and short-chain fatty acid content).

## 2. Materials and Methods

**Waste meals composition**. The meals (grapeseed, flaxseed and sea buckthorn meals) used for this study were purchased from S.C. OLEOMET-S.R.L., Bucuresti, Romania, in a dried form. Grapeseed flax and sea buckthorn meals were mixed in a ratio of 4:3:1 (GFS) based on their content in PUFA. The chemical composition (crude protein, crude fat, crude fibres and ash) of the GFS mixture was analysed by Weende's method according to the ISO methods (ASRO-SR EN ISO, 2010). Fatty acid composition of GFS and experimental diets were determined using a gas chromatograph (PerkinElmer, Clarus 500, Waltham, MA, USA) and a capillary chromatographic column for fatty acid methyl esters (BPX70, Agilent, Santa Clara, CA, USA) as already described [17].

**Experimental design.** Twelve cross-bred males TOPIG hybrid ((Landrace × Large White) × (Duroc × Pietrain)) piglets weaned at 27 days of age were provided by a private breeder (Fermeplus, Căzănești, Romania). The piglets had an average body weight of 11.25 ± 1.14 kg and were randomly distributed to one of the two experimental groups: the control group fed basic corn soybean diet (control diet) and an experimental group fed a diet with 10% mixture of grapeseed, flax and sea buckthorn meals in a ratio of 3:4:1 (GFS diet). The chemical composition of the GFS and of experimental diets as well as the calculated nutrient content of the experimental diets are presented in Tables 1 and 2. The diets were formulated to meet the NRC 2012 requirements for weaned pigs. The handling and protection of the animals used in this experiment were realised in accordance with the Romanian law 206/2004 and Council Directive 98/58/EC of the EU Council concerning the treatment of animals used for experimental purposes. The piglets were fed with the control or GFS diet for a period of 24 days. During the experiment, piglets had free access to solid feed and water. All animals appeared clinically normal, and no deaths resulted from the experiment.

**Table 1.** Chemical composition of meal mix (g/100 g sample).

| | GFS Mixture |
| --- | --- |
| Dry matter (DM) % | 93.19 |
| Crude protein (CP) % | 24.6 |
| Ether extract (EE) % | 11.68 |
| Crude fibre (CF) % | 19 |
| Ash % | 4.26 |

**Table 2.** Composition and calculated nutrient content of experimental diets (%).

| Items | Control Diet (%) | GFS Diet (%) |
| --- | --- | --- |
| Corn | 68.46 | 63.51 |
| Soybean meal | 19 | 14 |
| Flax meal | - | 5 |
| Grapeseed meal | - | 3.75 |
| Sea buckthorn meal | - | 1.25 |
| Corn gluten | 4 | 4 |
| Milk replacer | 5 | 5 |
| L-lysine-HCl | 0.31 | 0.40 |
| Methionine | 0.10 | 0.12 |
| Limestone | 1.57 | 1.58 |
| Monocalcium phosphate | 0.35 | 0.18 |
| NaCl | 0.10 | 0.10 |
| Choline premix | 0.10 | 0.10 |
| Phytase | 0.01 | 0.01 |
| Mineral–vitamin premix * | 1 | 1 |
| **Calculated nutrient content** | | |
| ME (kcal/kg) | 3282.60 | 3265.70 |
| Crude protein (%) | 18.70 | 18.29 |
| Lysine (%) | 1.20 | 1.20 |
| Met + Cys (%) | 0.72 | 0.72 |
| Calcium (%) | 0,90 | 0.90 |
| Phosphorus (%) | 0.72 | 0.72 |
| Fat (%) | 2.40 | 2.70 |
| Cellulose (%) | 4.24 | 4.90 |
| Dry matter | 89.44 | 89.49 |

* Mineral–vitamin premix (1%) was supplied per kg of diet as follows: vit. A 6000 IU, vit. D3 800 IU, vit. E 20 IU, vit. K1 1.0 mg, vit. B1 1.0 mg, vit. B2 3.0 mg, d-pantothenic acid 6.3 mg, niacin 10 mg, biotin 30 mg. vit. B12 20 mg, folic acid 0.3 mg, vit. B6 1.5 mg, Fe 80 mg, Zn 25 mg, Mn 30 mg, I 0.22 mg. Se 0.22 mg, Co 0.3 mg, antioxidants 60 mg and maize starch as the carrier.

After 24 days, all animals (6 animals/group) were slaughtered according to the EU Council directive 2010/63/CE and samples of the colon and colonic content were taken from each animal and stored at −80 °C until further analyses.

**Plasma biochemical parameters.** Blood samples were collected aseptically in sodium heparin tubes (Vacutest Kima, Arzergrande, Italy) from the left jugular vein of piglets on day 24 and then centrifuged at $800 \times g$ for 20 min to separate plasma as already described [18]. The following parameters: cholesterol, triglycerides, total protein, albumin, glucose, phosphorus, calcium, magnesium, iron, total bilirubin, creatinine, urea, activity of alanine aminotransferase (ALAT), aspartate aminotransferase (ASAT), alkaline phosphatise (AP) and gamma glutamyl transferase (GGT) were measured in plasma using a Clinical Chemistry benchtop analyser Horiba Medical—ABX Pentra 400 (Irvine, CA, USA) and corresponding reagents.

**Nitric oxide concentration.** The evaluation of the NO concentration in plasma was done using the Griess assay as already described in our previous study [19].

**Gene expression analysis**. The total RNA was extracted from colon samples using a Qiagen RNeasy minikit (Qiagen GmbH, Dusseldorf, Germany) and the concentration and quality of RNAs were evaluated using a bioanalyser (2100 Agilent Bioanalyzer, Agilent

Technologies, Santa Clara, CA, USA). An M-MuLV reverse transcriptase kit was used for the synthesis of the complementary DNA (Thermo Fischer Scientific, Waltham, MA, USA). qPCR (quantitative PCR) was used for the assessment of the gene expression of inflammatory cytokines: TNF-$\alpha$ (tumour necrosis factor alpha), interleukin 1 beta (IL-1$\beta$), interleukin 6 (IL-6) and interleukin 8 (IL-8) and junction proteins: claudin 4, claudin 7, occludin, zonula 1 (ZO-1), mucin 2 (Muc 2) and extracellular matrix protein (ECM1). The primer pairs (Eurogentec, USA) and the qPCR protocol were already described in our previous studies [20,21]. The changes in gene expression were assessed after the normalisation of qPCR data using two reference genes: cyclophilin A (CYP A) and glyceraldehyde-3-phosphate dehydrogenase (GAPDH) selected from a panel of six reference genes, using NormFinder software. Gene expression was computed using the 2($-\Delta\Delta$CT) method, and the results were expressed as relative fold change (Fc) compared with the control group.

**Cytokine production measurement.** The measurement of cytokines was done in colon samples homogenised in phosphate buffer containing 1% igepal, 0.5% sodium deoxycholate, 0.1% SDS and complete (EDTA-free) protease inhibitor cocktail tablets using the ELISA technique. Monoclonal antiporcine antibodies: IL-1$\beta$ (MAB6811), IL-8 (MAB5351) and TNF-$\alpha$ (MAB6902) (R&D Systems; Minneapolis, MN, USA) were used as the capture. The antibodies were used in conjunction with antiporcine cytokines—biotinylated antibodies: IL-1$\beta$ (BAF 681), IL-8 (BAF535) and TNF-$\alpha$ (BAF690). Streptavidin-HRP (Invitrogen, Camarillo, CA, USA) and TMB (tetramethylbenzidine) were used for the detection. The absorbance was read at 450 nm using a microplate reader (SUNRISE, TECAN, Grödig, Austria). Recombinant swine IL-1$\beta$, IL-8 and TNF-$\alpha$ were used as standards and the results were expressed as picograms of cytokine per millilitre, after normalisation to the total protein content of the samples.

**Immunoblotting Analysis.** The Western blotting method was used for the analysis of the expression levels of junction proteins (claudin-4, occludin and ZO-1). The equivalent of 30 mg protein of colon samples homogenised and lysed in RIPA buffer was separated on a 10% SDS-PAGE and transferred onto nitrocellulose membrane as already described by Pistol and co-workers [21]. The membranes were incubated for 2 h with rabbit anti-Claudin-4 and ZO-1 antibodies (Invitrogen, USA; diluted 1/1000), occludin and beta-actin antibodies (Cell Signaling Technology, Danvers, MA, USA; diluted 1/1000). After washing, the blots were incubated at room temperature for 1 h with a goat antirabbit antibody conjugated with horseradish peroxidase (dilution 1:2000, Cell Signaling Technology, Danvers, MA, USA). The antibody detection was performed as previously described [22].

**Microbial DNA extraction.** The microbial genetic material was extracted using the QIAamp DNA stool minikit extraction kit (QIAGEN, Dusseldorf, Germany), following the protocol provided by the manufacturer. The primer pairs (Eurogentec, Seraing, Belgium) and the PCR protocol for the amplification of universal 16S regions were already described in our previous study [23]. Serial dilutions of template DNA ($10^5$–$10^{12}$ molecules/$\mu$L) were used for the generation of standard curves. The quantification of the target bacteria genera DNA was performed using the RotorGene Q series systems using specific primer sequences for selected microorganism genus (Lactobacillus, Clostridium, Enterobacter, Prevotella and Bifidobacter) and PCR protocol as already described [23]. The sample fluorescence was measured at the last step of each cycle. The gene copy numbers of the target bacteria genera from the colonic digesta samples were determined by relating the Cq values to standard curves. The final copy number for each targeted bacteria genus was calculated using the following equation:

$$\text{No. bacterial copies digesta/grams} = (QM \times C \times DV)/S \times W$$

in which QM represents the quantitative mean of the copy number, C represents the DNA concentration of each sample, DV is the dilution volume of extracted DNA, S is the DNA amount (ng) which was used in the experiment and W stands for the weight of the colonic sample (g) (Metzler-Zebeli et al., 2013).

**Gas Chromatography method for SCFAs determination.** Short-chain fatty acids from the piglet colonic digesta samples were analysed in aqueous extracts (in a 1:2 weight per volume ratio), by gas chromatography as already described [23]. Briefly, after centrifugation, the aqueous extracts were injected on a gas chromatograph (Varian, 430-GC) fitted with an Elite-FFAP capillary column (inside diameter 320 mm) (Perkin Elmer, Waltham, MA, USA). A standard mix of volatile fatty acids were used as standards (CRM46975, Supelco, USA); the final results were expressed in µmol/g.

**Statistical analyses.** Differences between the two groups were analysed for significance using StatView software 6.0 (SAS Institute, Cary, NC, USA) with a one-way ANOVA followed by Fisher's PSLD test considering each pig as an experimental unit. $p$ values $< 0.05$ were considered to be statistically significant (* $p < 0.05$, ** $p < 0.01$, **** $p < 0.0001$). A principal component analysis (PCA) model was performed using Minitab 20.4 (Minitab, LLC, Pennsylvania) in order to analyse whether a combination of the selected markers of inflammations, epithelial integrity and microbiota could cluster GFS-treated animals separately from controls. Pearson's correlation coefficients between SCFA concentration and microbiota in the colon content were calculated using GraphPad Prism 8 (GraphPad Software, La Jolla, CA, USA).

## 3. Results

### 3.1. Chemical Composition of GFS and Diets

GFS contained a high concentration of PUFA (60.77 g/100 g FAME) in which 37.64 g/100 g FAME were omega-3 unsaturated fatty acids (Table 3).

**Table 3.** Fatty acids content of the meal mix and the diets.

| g FAME */100 g Total FAME | Fatty Acids | GFS Meal Mix | Control Diet | GFS Diet |
|---|---|---|---|---|
| Caprylic | C8:0 | 0.00 | 0.04 | 0.11 |
| Capric | C10:0 | 0.02 | 0.06 | 0.04 |
| Lauric | C12:0 | 0.03 | 0.06 | 0.03 |
| Myristic | C14:0 | 0.16 | 0.16 | 0.13 |
| Pentadecanoic | C15:0 | 0.11 | 0.32 | 0.87 |
| Pentadecenoic | C15:1 | 0.05 | 0.15 | 0.71 |
| Palmitic | C16:0 | 11.99 | 15.79 | 18.33 |
| Palmitoleic | C16:1 | 3.65 | 0.37 | 1.39 |
| Heptadecanoic | C17:0 | 0.08 | 0.13 | 0.03 |
| Heptadecenoic | C17:1 | 0.04 | 0.05 | 0.14 |
| Stearic | C18:0 | 2.99 | 3.26 | 4.59 |
| Oleic cis | C18:1 | 19.94 | 34.60 | 34.31 |
| Linoleic cis | C18:2n6 | 22.94 | 41.93 | 31.49 |
| Linolenic α | C18:3n3 | 37.36 | 1.19 | 6.22 |
| Octadecatetraenoic | C18:4n3 | 0.13 | 0.50 | 0.55 |
| Eicosadienoic | C20(2n6) | 0.04 | 0.28 | 0.36 |
| Eicosatrienoic | C20(3n6) | 0.03 | 0.05 | 0.06 |
| Eicosatrienoic | C20(3n3) | 0.07 | 0.07 | 0.14 |
| Arachidonic | C20(4n6) | 0.12 | 0.20 | 0.28 |
| Eicosapentaenoic | C20(5n3) | 0.08 | 0.14 | 0.00 |
| Lignoceric | C 24:0 | 0.08 | 0.26 | 0.00 |
| Nervonic C24 (1n9) | C24 (1n9) | 0.00 | 0.26 | 0.00 |
| Other fatty acids | | 0.11 | 0.12 | 0.21 |
| Saturated fatty acids (SFA) | | 15.45 | 20.09 | 24.13 |
| Unsaturated fatty acids (UFA) | | 84.44 | 82.46 | 75.65 |
| Monounsaturated fatty acids (MUFA) | | 23.68 | 38.10 | 36.55 |
| Polyunsaturated fatty acids (PUFA) | | 60.77 | 44.35 | 39.10 |
| n3 (omega-3) | | 37.64 | 1.90 | 6.91 |
| n6 (omega-6) | | 23.13 | 42.45 | 32.20 |
| TOTAL FATTY ACIDS | | 100 | 100 | 100 |

* FAME—fatty acids methyl esters.

When looking to the individual PUFA composition of GFS, it showed a high concentration of n-3 linolenic alpha acid (37.6%), n-6 linoleic acid (22.9%) and n-9 oleic acid (19.94%). The inclusion of the GFS mixture in the piglet's diet generated an important increase of the linolenic-$\alpha$ acid content (5.2 times compared with the control diet). This increase was also observed for other PUFA, but in lower proportion: eicosatrienoic (two-time increase), arachidonic acid (1.4-time increase). However, the oleic acid concentration in the GFS diet was not significantly different from the control, while a lower percentage of linoleic acid was present in the GFS diet compared to the control (31.49% vs. 41.93%).

### 3.2. GFS Effect on Animal Performance and Health Status

The animal performances of piglets fed with control or GFS diets for a period of 24 days are reported in Table 4.

**Table 4.** The effect of the diets on the performance of piglets.

| Parameters * | Control Diet | GFS Diet | *p* Value |
|---|---|---|---|
| Initial body weight, kg | 11.1 $\pm$ 1.17 | 11.3 $\pm$ 1.21 | 0.9385 |
| Final body weight, kg | 20.0 $\pm$ 3.22 | 23.5 $\pm$ 3.02 | 0.0792 |
| Average daily gain, kg | 0.384 $\pm$ 0.112 | 0.529 $\pm$ 0.140 | 0.1095 |
| Average daily feed intake, kg | 1.09 $\pm$ 0.158 | 1.00 $\pm$ 0.0501 | 0.8956 |
| Feed:gain ratio, kg:kg | 2.83 | 1.89 | 0.0681 |

* Data are presented as means $\pm$ standard error.

The GFS treatment tended to increase the final body weight (BW; 17.5% increase) and average daily gain (ADG, 37.7% increase), but had no significant effect on the average daily feed intake (ADFI). The animal performance tended to increase in the group fed the GFS diet.

The health status of the animals was investigated through the evaluation of serum biochemical parameters reflecting protein metabolism (total protein, albumin, total bilirubin, creatinine, urea), glucose metabolism (glycemia), lipide metabolism (cholesterol, triglycerides) as well as mineral status (phosphorus, calcium, magnesium, iron) and liver functionality (ALAT, ASAT, alkaline phosphatase, gamma glutamyl transferase) (Table 5).

**Table 5.** The effect of the diets on serum biochemical parameters of weaned piglets.

| Parameters * | Control Diet | GFS Diet | *p* Value |
|---|---|---|---|
| Glycemia, mg/dL | 128 $\pm$ 26.1 | 132.6 $\pm$ 23.3 | 0.7855 |
| Cholesterol, mg/dL | 111 $\pm$ 17.4 | 104.9 $\pm$ 10.2 | 0.1265 |
| Triglycerides, mg/dL | 47.7 $\pm$ 6.92 | 57.5 $\pm$ 5.23 | 0.0721 |
| Total protein, mg/dL | 6.07 $\pm$ 0.252 | 5.90 $\pm$ 0.235 | 0.2898 |
| Albumin, g/dL | 3.30 $\pm$ 0.511 | 3.00 $\pm$ 0.0216 | 0.1449 |
| Total bilirubin, g/dL | 0.171 $\pm$ 0.0232 | 0.141 $\pm$ 0.0209 | 0.0875 |
| Creatinine, mg/dL | 1.50 $\pm$ 0.423 | 1.10 $\pm$ 0.352 | 0.1367 |
| Urea, mg/dL | 28.1 $\pm$ 5.80 | 27.6 $\pm$ 3.10 | 0.8444 |
| Phosphorus, mg/dL | 7.32 $\pm$ 1.23 | 6.79 $\pm$ 0.629 | 0.3438 |
| Calcium, mg/dL | 12.6 $\pm$ 0.914 | 12.8 $\pm$ 1.32 | 0.7670 |
| Magnesium, mg/dL | 1.60 $\pm$ 0.312 | 1.65 $\pm$ 0.159 | 0.7525 |
| Iron, µg/dL | 134 $\pm$ 40.1 | 93.2 $\pm$ 10.1 | 0.0680 |
| ALAT (TGP), U/L | 61.2 $\pm$ 12.5 | 58.7 $\pm$ 7.06 | 0.6977 |
| ASAT (TGO), U/L | 42.8 $\pm$ 6.90 | 41.3 $\pm$ 9.92 | 0.7677 |
| Alkaline phosphatase (ALK), U/L | 273 $\pm$ 52.1 | 321 $\pm$ 53.2 | 0.1530 |
| Gama glutamyl transferase (GGT), U/L | 17.5 $\pm$ 1.72 | 23.7 $\pm$ 2.86 | 0.0698 |

* Data are presented as means $\pm$ standard error.

No significant difference was observed between the two groups and all parameters were in the normal range area for this category of animal. Moreover, the effect of the GFS treatment on total immunoglobulins synthesis was assessed as an important indicator of the capacity of the organism to respond to external threats. Figure 1A shows no significant

differences between the two treatments for IgG ($p = 0.197$), IgA ($p = 0.377$) or IgM ($p = 0.160$). However, the assessment of nitric oxide in the plasma of the weaned piglets, as an important marker of nitrosative stress, showed a significant decrease in the GFS group from 9.82 to 7.99 ($p = 0.024$) compared to the control group (Figure 1B).

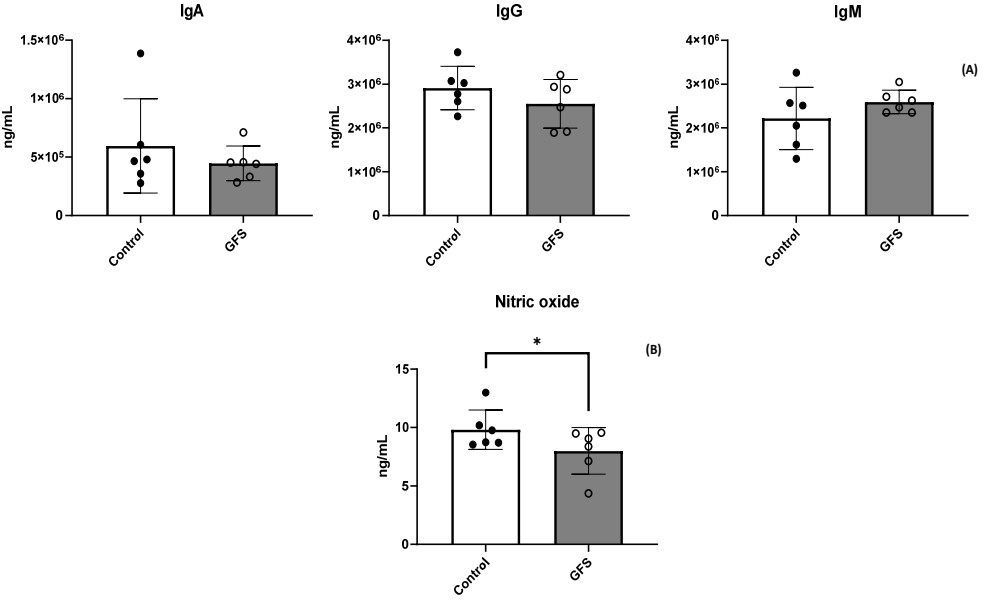

**Figure 1.** Effect of GFS diet on selected immune markers in the plasma of weaned piglets. (**A**) Concentration of immunoglobulins IgA, Ig G and Ig A in the plasma of control (white columns) and GFS-treated (grey columns) piglets. (**B**) Concentration of nitric oxide in the plasma of control (white columns) and GFS-treated (grey columns) piglets. The results are displayed as mean ± standard error (SE). * indicates a significant difference between control and GFS-treated piglets ($p < 0.05$).

### 3.3. GFS Effect on Proinflammatory Cytokines

In order to investigate the capacity of the GFS diet to improve the inflammatory response at the intestinal level in weaned piglets, the effect of GFS on proinflammatory cytokine was investigated both at the gene and protein level. The presence of GFS in the diet significantly reduced the expression of the IL-1 beta gene ($p = 0.035$, Figure 2A). The expression of other proinflammatory cytokines gene was always under the control values, but this effect was not significant (Figure 2A). At the protein level, the GFS treatment was able to significantly reduce by 1.78 times the concentration of IL-1 beta ($p < 0.0001$) and by 1.37 times the concentration of TNF-alpha ($p = 0.05$) in the colon of the piglets fed the GFS diet, while no effect of GFS was observed for the IL-8 chemokine concentration (Figure 2B).

### 3.4. GFS Effect on Junction Proteins, Mucins and Proteins of Extracellular Matrix

The effect on epithelial integrity of the colon was investigated through the assessment of the expression of junction proteins both at the gene and protein level.

The GFS diet significantly increased the gene expression of occludin (a 1.94-time increase, $p = 0.0168$), claudin 7 (1.49-time increase, $p = 0.467$) and extracellular protein matrix (2.18-time increase, $p = 0.05$) compared to the control diet (Figure 3A).

The fold change in the gene expression of the other junction proteins was always numerically higher than in the control group, but not statistically significant. The western blot analyses of junction proteins expression indicated that the presence of GFS in the diet significantly increased the claudin 4 protein expression ($p = 0.035$) compared to that of the control (3.11 A.U. vs. 2.3 A.U.). The expression of the two other junction proteins investigated (occludin and ZO-1) was not modified by the presence of GFS in the diet (Figure 3B).

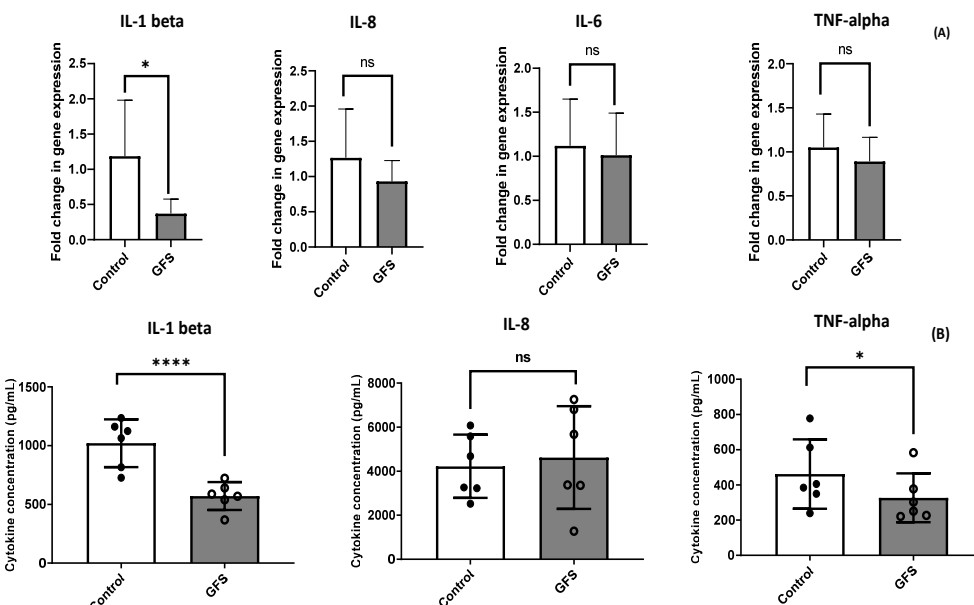

**Figure 2.** Effect of GFS diet on proinflammatory cytokines in the colon of weaned piglets. (**A**) Fold change in gene expression of proinflammatory cytokines IL-1 beta, IL-8, IL-6 and TNF-alpha in the colon of control (white columns) and GFS-treated (grey columns) piglets. (**B**) Concentration of proinflammatory cytokines IL-1 beta, IL-8 and TNF-alpha in the colon of control (white columns) and GFS-treated (grey columns) piglets. The results are displayed as mean ± standard error (SE). * indicates a significant difference between control and GFS-treated piglets ($p < 0.05$). **** indicates a highly significant difference between control and GFS-treated piglets ($p < 0.0001$). ns indicates nonsignificant difference.

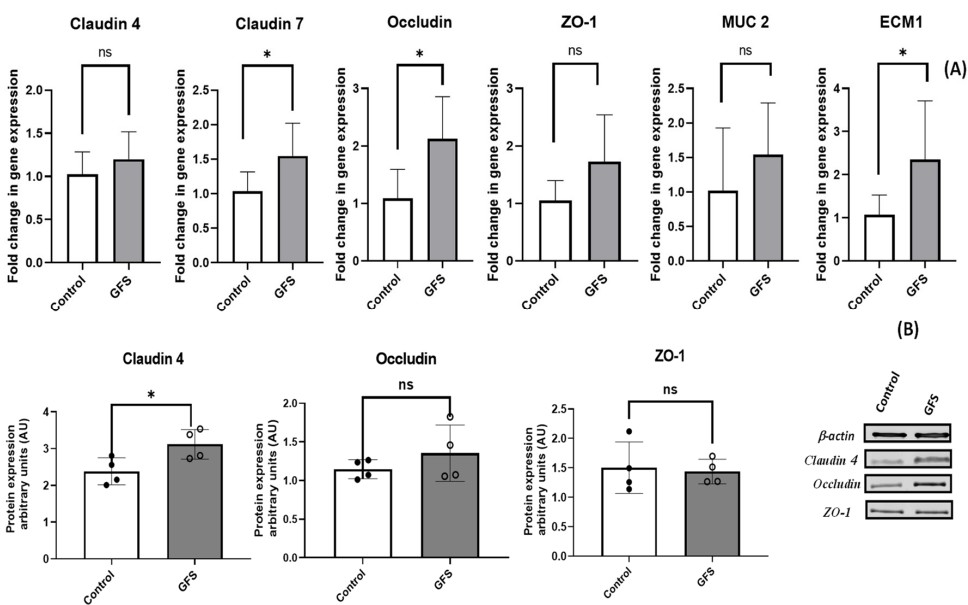

**Figure 3.** Effect of GFS diet on junction proteins in the colon of weaned piglets. (**A**) Fold change in gene expression of junction proteins (claudin 4, claudin 7, occludin, ZO-1) mucins (MUC 2) and proteins of extracellular matrix (ECM1) in the colon of control (white columns) and GFS-treated (grey columns) piglets. (**B**) Junction protein expression (claudin 4, occludin, ZO-1) in the colon of control (white columns) and GFS-treated (grey columns) piglets; Western blot images of junction proteins and beta-actin in the colon of control and GFS-treated piglets. The results are displayed as mean ± standard error (SE). * indicates a significant difference between control and GFS-treated piglets ($p < 0.05$). ns indicates nonsignificant difference.

### 3.5. GFS Effect on Selected Microbiota Populations and on Short-Chain Fatty Acids

In order to determine the effect of the GFS diet on the intestinal microbiota, the absolute abundance of five key microbial strains (Clostridium, Bifidobacterium, Enterobacter, Prevotella and Lactobacillus) relevant for the gut health were investigated in the colonic content of both control and GFS-treated piglets (Figure 4A–E).

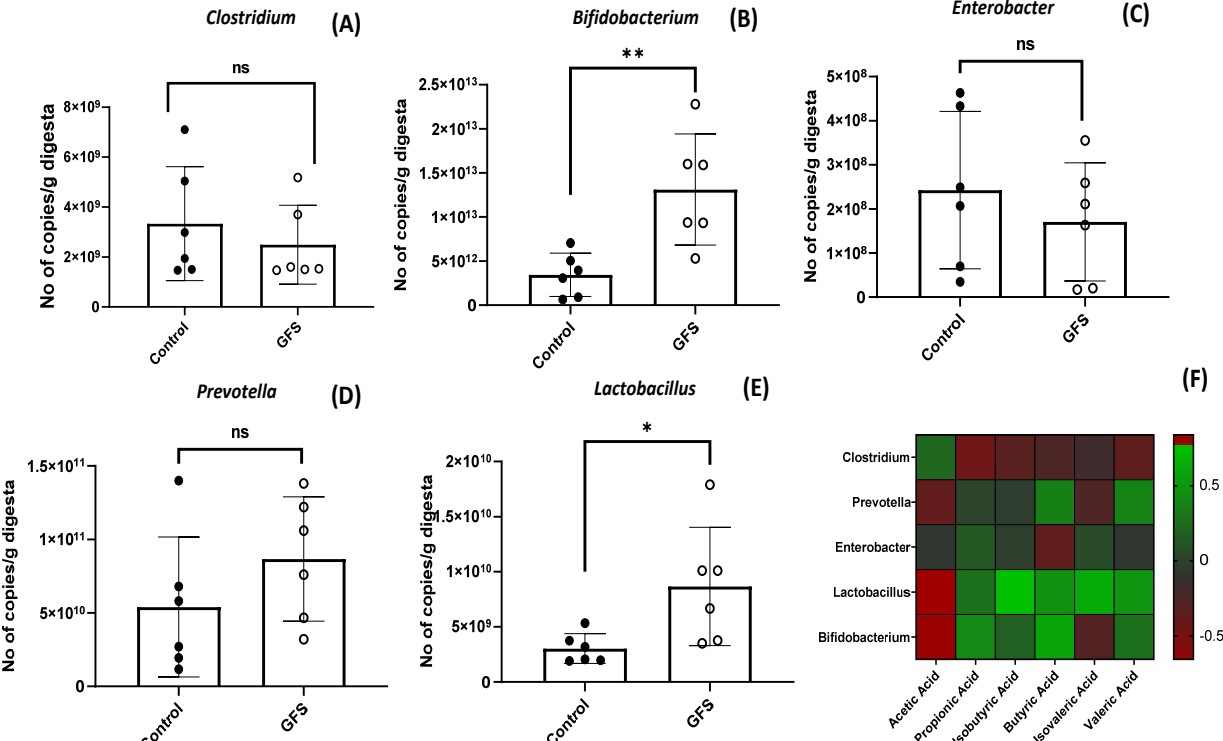

**Figure 4.** Effect of GFS diet on selected microbiota species and on short-chain fatty acid synthesis (**A–E**) Bacterial absolute abundances for targeted genera. (**F**) Heatmap representing Pearson's correlations between SCFA concentration and genera of the microbial community in the colonic content of piglets fed a control and GFS-enriched diet (N = 12). Green indicates positive correlation; red indicates negative correlation. The results are displayed as mean ± standard error (SE). * indicates a significant difference between control and GFS-treated piglets ($p < 0.05$). ** indicates a very significant difference between control and GFS-treated piglets ($p < 0.01$).

The presence of GFS in the diet had a positive effect on the *Bifidobacterium* and *Lactobacillus* species, consisting in an increase of the microbial strain abundance by 3.8 times ($p = 0.0057$) and by 2.85 times ($p = 0.0316$), respectively, compared to that of the control diet. A slight increase was also observed for Prevotella ($p = 0.234$), while no significant effect was noticed for *Clostridium* and *Enterobacter*. Table 6 shows the short-chain fatty acids concentration in the colonic digesta collected from both control and GFS-treated piglets. The GFS diet was responsible for a significant decrease in acetic acid by 22.5% and a significant increase of propionic and butyric acids by 23.3% and 62%, respectively, compared to the control diet. Moreover, a tendency to increase the concentration of valeric acid ($p = 0.08$) was observed in the colonic content of the GFS group compared to the control group, while no difference was noticed between the two groups for the isobutyric and isovaleric acids.

**Table 6.** Short-chain fatty acids content in colonic digesta collected from piglets.

| SCFA/Exp Group * | Acetic Acid | Propionic Acid | Isobutyric Acid | Butyric Acid | Isovaleric Acid | Valeric Acid |
|---|---|---|---|---|---|---|
| Control | 61.33 ± 1.29 | 20.16 ± 0.78 | 1.8 ± 0.12 | 11.72 ± 0.41 | 1.88 ± 0.18 | 3.08 ± 0.22 |
| GFS | 47.48 ± 0.74 | 24.82 ± 1.21 | 2.34 ± 0.2 | 19.1 ± 0.72 | 2.12 ± 0.46 | 4.11 ± 0.19 |
| *p* value | <0.0001 | <0.01 | 0.27 | <0.0001 | 0.9 | 0.08 |

* Data are presented as means ± standard error.

The results of Pearson's correlations between the SCFA ratios and microbiota data of the colon are shown in the heatmap from the Figure 4F. *Lactobacillus* and *Bifidobacterium* were positively correlated with butyrate production (r = 0.56 and r = 0.68). Positive correlations were observed also between *Lactobacillus* and isobutyric (r = 0.83), isovaleric (r = 0.71) and valeric (r = 0.57) acids, while *Bifidobacterium* was positively correlated with propionic acid concentration (r = 0.51).

*3.6. Data Clustering Using Principal Component Analysis*

Further, we used a principal component analysis (PCA) inputting values of the investigated markers of inflammation, microbiota and junction proteins in order to test if the control and GFS groups could be separated into two independent clusters, and which variables would influence such separation (Figure 5).

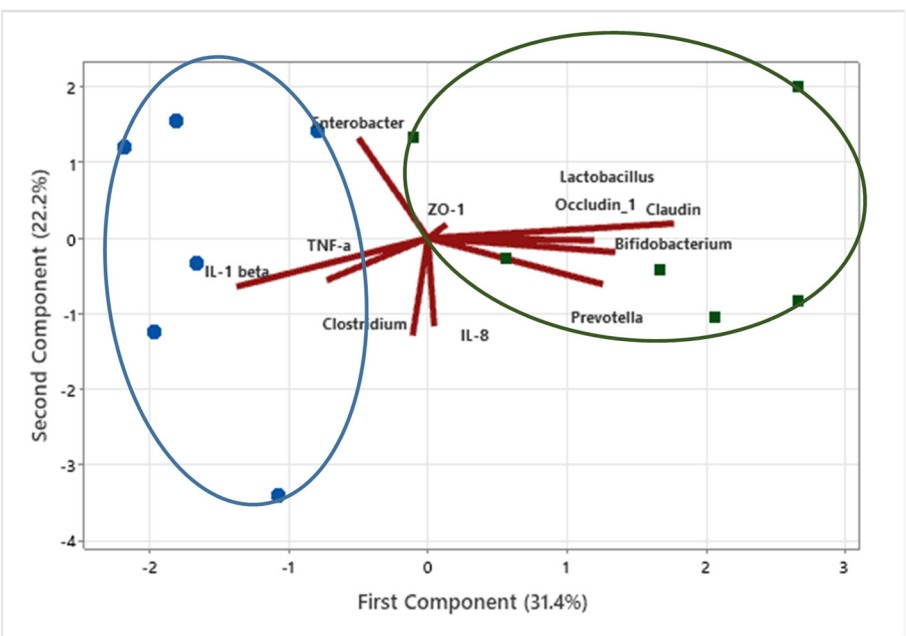

**Figure 5.** Estimation of the relationships between inflammation, microbiota and junction proteins in the colon samples using principal component analysis.

The PCA analysis expressed the correlation between selected markers of inflammations, epithelial integrity and microbiota under the action of the two dietary treatments. Each point represents one animal (control (blue), GFS (green)). A biplot rays vector analysis (in red) was used to show the impact of each marker in the data distribution of the PCA model.

The sum of principal components PC1 and PC2 explained 53.6% of the variations among treatments and two distinct clusters for the control and GFS diets were identified based on the PCA. The vector analysis integrated with a PCA model showed that the increase of bacterial abundance for *Lactobacillus*, *Bifidobacterium* and *Prevotella* species as well as of the expression of junction proteins occludin and claudin were more associated with GFS-treated piglets, whereas the expression of proinflammatory cytokines IL-1 beta

and TNF-alpha were associated with the control group. This finding highlights a potential beneficial role of the GFS treatment in the colon of weaned piglets.

## 4. Discussion

Polyunsaturated fatty acids are important constituents of the cell membranes, responsible for the maintenance of the membrane fluidity and the regulation of intracellular signalling processes, gene expression and production of cell mediators [24]. A change of dietary PUFA intake influences the structure of the cell membrane in different tissues, including the gut of piglets [25]. At the gut level, PUFA play a major role in relation to epithelial barrier functions, mucosal immune responses, inflammation and oxidative stress [26]. The essential fatty acids (linoleic acid and α-linolenic acid) cannot be synthesized in the animal organism and should be taken from different dietary sources [27]. Several studies have shown that the inclusion of PUFA sources in the diet of weaned piglets may be used as a strategic tool to improve the performances of the animals, as well as their gut health and function [28]. The present study was focused on the assessment of the efficacy of a diet including a mix of different agro-industrial waste (flaxseed, grapeseed and sea blackthorn) rich in omega 3 PUFA, to improve the piglets' health and to counteract the negative effects associated with the weaning stress. Several studies have shown that feeding piglets with diets rich in omega 3 PUFA resulted into a quicker adaptation to the rapid change of a diet after weaning [29]. However, few studies have investigated the effect of dietary omega 3 PUFA on gut health, microbiota composition and on intestinal immune response, including inflammation in weaned piglets. First of all, we investigated the effect of PUFA-enriched diet on weaned piglets' performance. The inclusion of a 10% mixture of grapeseed, flax and sea buckthorn meals in the diet of weanling piglets tended to increase the final body weight ($p = 0.080$) and average daily gain (37.7% increase; $p = 0.079$) and did not significantly improve the average daily feed intake (ADFI) or feed-to-growth ratio (F:G) in weaned pigs. Our findings are consistent with a previous study of Li and co-workers who showed that the administration of a diet supplemented with 3% n-3 PUFA to weanling piglets did not result in a significant improvement of ADG, ADFI or G:F ratio [29]. However, PUFA supplementation to other categories of pigs resulted in significant improvement of the BW, ADG and ADFI [30]. Similarly, literature data have shown that in general the supplementation of grower and finisher feed with fatty acids increased the growth speed, reduced the feed intake and improved the gain efficiency; however, beneficial effects on stress were not observed in piglets during the weaning period [31]. To test the hypothesis that fats contained in dietary waste might improve the gut health of the weaned piglets, we investigated the GFS diet effect on inflammation, gut epithelium integrity, selected microbiota populations and short-chain fatty acids synthesis. Besides its major role in the digestion, absorption and metabolism of dietary nutrients, the gastrointestinal tract is the largest immunological organ in the body, around 70–80% of the immune cells being located in the gut [32]. Weaning stress is characterised by biochemical and histological changes that occur in the small intestine and that cause an exacerbation of the secretion of proinflammatory cytokines and induce severe intestinal inflammation [33]. Weaned piglets have an immature gut and an immature immune system; thus, they are very vulnerable to pathogen infection and weaning stress [3]. The inclusion of dietary fat sources in the weaned pigs feed may be used to reduce inflammation and other negative effects associated with these conditions. Omega-3 PUFA were described as molecules with anti-inflammatory properties in different species [34–36], including swine [37,38]. Indeed, in our study, the inclusion of agro-industrial waste rich in PUFA into the weaned piglets' diet significantly reduced the concentration of proinflammatory cytokines IL-1 beta and TNF-alpha. Furthermore, the supplementation of the diets for sows and piglets with a natural feed ingredient containing 90% PUFA significantly lowered the concentration of other proinflammatory marker—myeloperoxidase—on the faeces of postweaning piglets [39]. Literature studies have shown that the inhibition of the NF-κB signalling pathway involved in regulation of the innate and adaptive immune functions and inflammatory responses is one of the

mechanisms that PUFA trigger to reduce the proinflammatory markers [40,41]. The proinflammatory cytokines play a critical role in the alteration of intestinal barrier function [42]. For example, TNF-$\alpha$ impairs endothelial barrier function [43] by the alteration of the tight junction proteins [44]. Tight junctions (TJ) are intercellular adhesion complexes composed of transmembrane proteins (claudins), TJ-associated marvel proteins (e.g., occludin), junctional adhesion molecules (JAM) and cytosolic proteins (e.g., zonula), with a role in the connection of transmembrane components to the cytoskeleton [45]. Dietary omega-3 PUFA can protect gut barrier function against injury induced by different stress factors. Recent studies have shown that weaning stress is associated with an increase of intestinal permeability through a decrease of the tight junction proteins in piglets [46]. The expression of junction proteins zonula 1, claudin-1, claudin-5 and claudin-8 was also increased after omega-3 PUFAs administration in rats with experimentally induced colitis [47]. In the same study, omega-3 PUFAs improved the histological score and ameliorated the disruption of the tight junction structure. Our study also showed that GFS administration increased the expression of junction proteins claudin and occludin and of the extracellular protein matrix. The weaning period is a very important stage in the pig's life, as during weaning, the animals are very susceptible to pathogens and develop postweaning diarrhoea [48]. The gut microbiota contribute to the enhanced resistance of piglets against pathogenic bacteria and have a major contribution to the immune system development with an impact on the health of piglets [49]. The shift in the gut microbiota profile results from the abrupt transition from liquid feed to a solid diet that modifies the substrate availability in the intestine [50]. A comparison between the microbiome profile of healthy and piglets with diarrhoea has shown a reduced number of *Lactobacillus* bacteria, as well as an increased abundance of pathogenic bacteria *Escherichia* and *Shigella* [51]. The ingestion of diets rich in PUFA is able to modify the gut microbiome and reduce intestinal dysbiosis [52] and the inflammation in the colon by targeting the bacterial diversity of colonic microbiota [53]. Our results showed an increase abundance of the Lactobacillus and Bifidobacterium species after the administration of the diet with a high level of omega-3 fatty acids; these bacteria exert positive health benefits on their host as they are able to reduce the diarrhoea incidence when they are administered as probiotics [54]. The administration of the diet enriched in omega 3-PUFA to the piglets also resulted in an increase of the abundance of the *Prevotella*; the relative abundance of *Prevotella* increases was also observed in weaned piglets with introduction of a plant-based diet [49]. Similarly, other studies have found that the gut microbiota of healthy piglets had a higher abundance of *Prevotellaceae* and Lactobacillaceae compared to diarrheic piglets [55]. Short-chain fatty acids (SCFA) are produced via microbial fermentation of dietary carbohydrates by the intestinal microbiota and are important for maintaining intestinal homeostasis [56]. Weaning significantly changes the colonic SCFA concentration and consequently the growth and development of colon [57]. In our study, the GFS diet affected the SCFA synthesis by decreasing the concentration of acetic acid and increasing the concentration of propionic and butyric acids. SCFA represent an important energy source for colonocytes and they are essential for the prevention of intestinal diseases, the proliferation of GIT tissue and the absorption of minerals [58]. In particular, butyrate reduces intestinal inflammation as well as systemic infection, and it acts as an important modulator of the microbiome, metabolic pathways regulator and antioxidant [59].

## 5. Conclusions

The inclusion of a combination of agro-industrial wastes (grapeseed, flaxseed and sea blackthorn meals) in a ratio of 3:4:1 in the diet of weaned piglets increased the content of omega-3 polyunsaturated fatty acids. Our data showed that the administration of a GFS diet positively influenced the gastro-intestinal health of piglets after weaning through a decrease of inflammation, an improvement of the gastro-intestinal barrier, modulation of microbiota and short chain fatty acids synthesis. Taken together, these results show that agro-industrial wastes rich in omega-3 PUFA can be used as an ecological and environmen-

tally friendly nutritional intervention for improving the negative effects associated with the weaning stress.

**Author Contributions:** Conceptualisation, D.E.M. and I.T.; microbiota analysis, I.G.; short fatty acids composition, A.E.C.; Western blot analyses, biochemistry, antibody and cytokine assessment G.C.P. and A.C.A.; qPCR analyses, C.V.B. and G.C.P.; data analysis and statistics, D.E.M.; writing—original draft preparation, D.E.M.; writing—review and editing, I.T.; funding acquisition, D.E.M. and I.T. All authors have read and agreed to the published version of the manuscript.

**Funding:** This work was financed through the projects PED 396/2019 and 8PFE/2021 financed by Romanian Ministry of Research, Innovation and Digitalization.

**Institutional Review Board Statement:** The animal study protocol was approved by Ethics Committee of National Research and Development Institute for Biology and Animal Nutrition Balotesti Romania (protocol code 356/9.04.2021).

**Informed Consent Statement:** Not applicable.

**Data Availability Statement:** The study did not report any data.

**Conflicts of Interest:** The authors declare no conflict of interest. The funders had no role in the design of the study; in the collection, analyses, or interpretation of data; in the writing of the manuscript; or in the decision to publish the results.

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
