# Peer review of "The Use of Agro-Industrial Waste Rich in Omega-3 PUFA during the Weaning Stress Improves the Gut Health of Weaned Piglets"

_agriculture, doi:10.3390/agriculture12081142_

Round 1

Reviewer 1 Report

This article explored the potential possibility of agro-industrial wastes rich in omega-3 PUFA in preventing weaning stress. It has practical production significance in agriculture and environmental protection. Followings are some suggestions in details.

1. P1, it well be better if can add some descriptions about the production status of those waste.

2. P2, line 54-56, please list the reference in detail.

3. P6, are there any differences in diarrhea index or the severity of diarrhea between two groups? If so, it can support your view well because the diarrhea is one of the important symptoms of weaning stress. Refer to 'Chunyan Xie, et al. Enteromorpha polysaccharide-zinc replacing prophylactic antibiotics contributes to improving gut health of weaned piglets. Animal Nutrition, 2021, 7(3): 641-649'.

4. P6, line 194-196, “GFS effect on animal performance and health status” is a level 3 heading, the explanation should be mentioned in the table captions or in the method.

5. P6, table 3, there is a great change in average daily gain and feed:contain ratio, please check the statistical method.

6. P6, line 208-212, it’s better to be written in the discussion.

7. P8, figure 3, “(A)” is missing, please modify.

8. P9, the problem is the same as figure 3, please modify.

9. P9, figure 4 (F), the positive correlation should be modified to green in the legend.

Author Response

This article explored the potential possibility of agro-industrial wastes rich in omega-3 PUFA in preventing weaning stress. It has practical production significance in agriculture and environmental protection. Followings are some suggestions in details.

The Authors wish to thanks the Reviewer for the appreciations concerning the quality of the manuscript and for the comments and the suggestion that will improve considerably the final form of the manuscript.

Reviewer. P1, it well be better if can add some descriptions about the production status of those waste.

Authors: Thank you for this comment. The total number of publications available describing processing the production status of those waste is rather limited and despite our literature research we cannot find any value data concerning the worldwide amount of these wastes/year. We can only presume that the disponible amount of these wastes is very high as we know that large surfaces or land are cultivated with flaxseed, grape and sea buckthorn.

Concerning your observation, we have realized a literature search and we can provide you some information that can provide a general view about the potential for waste generation of flaxseed, grape and sea buckthorn

  • For flax: In 2016, world area seeded to flax was 2.76 million hectares producing 2.93 million tons of flaxseed based on the available data (FAOSTAT, 2018); the oil recovery is considered to be between 70.1–85.7 %
  • For grape: Worldwide production of grape was 77,137,016 tons of grape/year (https://www.atlasbig.com/en-us/countries-grape-production); grape seeds representing 6% of total solid waste generated by winery (Baroi et al. 2022)
  • For sea buckthorn, we were no able to find available data concerning the worldwide production. However there are some papers that have shown sea buckthorn production in different countries: ex Mongolia 1,100-1,500t/year (Gonchigsumlaa et 2020); the yield percentage for juice extraction is about 70% (Cenkowski et al. 2006). The yield percentage for seed oil extraction is approximately 12% whereas the peel and the pulp give out an approximate yield percentage value of 6% (Dulf 2012).

Thus, even it looks like the amount of wastes generated each year is very high we cannot provide any literature value data for each waste in particular. Maybe in the future if the use of the wastes will be more important these kinds of data will be available

Reviewer . P2, line 54-56, please list the reference in detail.

Authors: Thank you for this comment. As suggested the references were now added for this paragraph in the new version of the manuscript.

Reviewer .  P6, are there any differences in diarrhea index or the severity of diarrhea between two groups? If so, it can support your view well because the diarrhea is one of the important symptoms of weaning stress. Refer to 'Chunyan Xie, et al. Enteromorpha polysaccharide-zinc replacing prophylactic antibiotics contributes to improving gut health of weaned piglets. Animal Nutrition, 2021, 7(3): 641-649'.

Authors: Thank you for this comment. Indeed, diarrhea is one of the important symptoms of weaning stress. In our experiment we didn’t notice any diarrhea, probably due to the fact that our farm is not a commercial farm but an experimental one, were the animals have better conditions than in commercial farms

Reviewer . P6, line 194-196, “GFS effect on animal performance and health status” is a level 3 heading, the explanation should be mentioned in the table captions or in the method.

Authors: Thank you for this suggestion. The explanation was now mentioned in the M&M section in the new version of the manuscript.

Reviewer . P6, table 3, there is a great change in average daily gain and feed: contain ratio, please check the statistical method.

Authors: Thank you for this comment. We have verified once again the statistics for feed: gain ratio, no statistical difference was observed. P values are now included in the table 4 (former table 3).  

Reviewer . P6, line 208-212, it’s better to be written in the discussion.

Authors: Thank you for this comment. The suggested change was done.

“The inclusion of 10% mixture of grape seed, flax and sea buckthorn meals in the diet of weanling piglet’s tended to increase the final body weight (p=0.080) and average daily gain (37.7% increase; p = 0.079) and did not significantly improve average daily feed intake (ADFI) or growth to feed ratio (F: G) in weaned pigs.”

Reviewer . P8, figure 3, “(A)” is missing, please modify.

Authors: Thank you for this observation. We have added the missing letter

Reviewer . P9, the problem is the same as figure 3, please modify.

Authors: Thank you for this observation. We have added the missing letter

Reviewer . P9, figure 4 (F), the positive correlation should be modified to green in the legend.

Authors: Thank you for this observation. We have done the modification in the legend

Reviewer 2 Report

The authors of the Ms entitled “ The use of agro-industrial waste rich in omega-3 PUFA during the weaning stress improve the gut health of weaned piglets” tested the hypothesis, GFS waste can counteract weaning stress and to improve piglets’ gut health by using a nutritional intervention, consisting in a mix of agro-industrial wastes (grapeseed, flaxseed, and sea blackthorn meals) rich in omega-3 PUFA. The authors used a twelve cross-bred TOPIG hybrid piglets 17 with an average body weight of 11.25 kg were randomly distributed to one of the two experimental groups, control vs. GFS goup. The authors concluded that agro-industrial wastes rich in omega-3 PUFA can be used as an ecological, environmentally friendly nutritional intervention for improving the negative effects associated with the weaning stress. This is a very well thought, presented and discussed  research that deserve publication in Q1  journal.

    My comments are in the following:

1. The abstract section L 16, plz reorder flaxseed, grapeseed, and sea blackthorn meals as grapeseed, flaxseed, and sea blackthorn meals to fits the abbreviation used there after as GFS

2. In the introduction section,the novelty and/or the added value of this work should be focus

3. The basis of choosing the these three wastes  and  percent of combination (3, 4,1) should be justified, L 72.

4. L 85, Plz check if the second in is necessary “realized in in”

5. I suggest to present the chemical composition of GFS in a table .

6. The from of feed should be added in L 88.

7. L 90, detailed information regrading blood samples were collection should be declared.

8. Plz add number of samples used for the carried-out measurements (n=???).

9. In the statistical analysis section, plz add the experimental unit used, also add if any transformation were done for data in %.

10. Please add the P value in tables 3&4.

11. The numbers in tables should be following this rule: xxxx, xxx, xx.x, x.xx, 0.xxx, 0.0xxx.

12.  In the conclusion section, L 416&417 Plz, reorder flaxseed, grapeseed, and sea blackthorn meals as grapeseed, flaxseed, and sea blackthorn meals to fits the abbreviation used there after as GFS

13.  L 418, in the conclusion section  Plz add the dose and ratios of GFS for taken home massage.

14.  In the conclusion section, don’t repeat the in conclusion, the title is adequate. 

Author Response

The Authors wish to thanks the Reviewer for the appreciations concerning the quality of the manuscript and for the comments and the suggestion that will improve considerably the final form of the manuscript.

Reviewer 1

Reviewer 1. The abstract section L 16, plz reorder flaxseed, grapeseed, and sea blackthorn meals as grapeseed, flaxseed, and sea blackthorn meals to fits the abbreviation used there after as GFS.

Authors: Thank you for this suggestion. We have modified the abstract accordingly.

Reviewer 1. In the introduction section,the novelty and/or the added value of this work should be focus

Authors: Thank you for this suggestion. The novelty of the study was added to the introduction section in the new version of the manuscript:

 “We presume that the new formula destinated to weanling piglets will provide for animals a wide diversity of PUFA from different sources (flaxseed, grapeseed and sea blackthorn meals) that will assure an easily transition of the piglets through the weaning period “

Reviewer 1. The basis of choosing these three wastes and percent of combination (3, 4,1) should be justified, L 72.

Authors: Thank you for your comment. The grape seed, flaxseed and sea buckthorn meals were chosen on the basis of their rich content in bioactive compounds (PUFA) and in order to assure a higher diversity of PUFA than in the case of the use of a single meal. The flaxseed contains mainly n-3 PUFA (eg C18:3 – 50-60% of the total fatty acids), grape seed contain in principal n-6 PUFA (65-75% C18:2) and sea buckthorn seed contain both n-3 (24%) and n-6 FA (34%). In the mixture flaxseed was introduced in the highest amount as it contains the highest amount of n-3 known for its anti-inflammatory properties. Beside their content in PUFA, the other two meals could bring other bioactive compounds to the diet with antioxidant activity. For example, grape seeds contain mainly flavonoids, including gallic acid, the monomeric flavan-3-ols catechin, epicatechin, gallocatechin, epigallocatechin, and epicatechin 3-O-gallate, and procyanidin dimers, trimers, and more highly polymerized procyanidins. Also, sea buckthorn contains polyphenols (epicatechin, epigallocatechin, gallic acid, proanthocyanidins, chloregenic acid) and flavonoids (quercetin, isorhamnetin, kampferol glycosides, lutoelin, myricetin). These compounds can have anti-oxidant properties beside the anti-inflammatory properties of the PUFA, already demonstrated in the present paper, and can provide extra benefits for the animal health.

The following sentence was added in the M&M section:

“Grape seed flax, and sea buckthorn meals were mixed in a ratio of 4:3:1 (GFS) based on their content in PUFA”.

Reviewer 1. L 85, Plz check if the second in is necessary “realized in in”

Authors: Thank you for this observation. We have made the correction in the new version of the manuscript.

Reviewer 1. I suggest to present the chemical composition of GFS in a table.

Authors: Thank you for this suggestion. We have presented the information concerning chemical composition of GFS into a table that can be found now in the new version of the paper.

Also, the following sentence was added in the M&M section:

“Chemical composition of GFS and of experimental diets as well as the calculated nutrient content of experimental diets are presented in Table 1 and Table 2.”

Reviewer 1. The from of feed should be added in L 88.

Authors: Thank you for this observation. Feed in solid form was used. We have added this information in M&M section.

Reviewer 1. L 90, detailed information regrading blood samples were collection should be declared.

Authors: Thank you for this suggestion. We have added this information in M&M section.

“Blood samples were collected aseptically in sodium heparin tubes (Vacutest Kima, Arzergrande, Italy) from left jugular vein of piglets on day 24 and then centrifuged at 800 g for 20 min to separate plasma as already described (16)”

Reviewer 1. Plz add number of samples used for the carried-out measurements (n=???).

Authors: Thank you for this comment. The samples were collected from all twelve animals (6 animals/group). This information was now added in the M&M section, in the new version of the manuscript.

“After 24 days, all animals (6 animals/group) were slaughtered according to the EU Council directive 2010/63/CE and samples of colon and colonic content were taken from each animal and stored at -800C until further analyses”.

Reviewer 1. In the statistical analysis section, plz add the experimental unit used, also add if any transformation were done for data in %.

Authors: Thank you for this suggestion. Each pig was considered an experimental unit. This information was included in the Statistical analysis section.

Differences between the two groups were analysed for significance using StatView software 6.0 (SAS Institute, Cary, NC) with one-way ANOVA followed by Fisher PSLD test considering each pig as an experimental unit.

Reviewer 1. Please add the P value in tables 3&4.

Authors: Thank you for this suggestion. The p values were added as suggested in the new version of the manuscript.

Reviewer 1. The numbers in tables should be following this rule: xxxx, xxx, xx.x, x.xx, 0.xxx, 0.0xxx.

Authors: Thank you for this observation. The values in the tables were modified as suggested.

Reviewer 1. In the conclusion section, L 416&417 Plz, reorder flaxseed, grapeseed, and sea blackthorn meals as grapeseed, flaxseed, and sea blackthorn meals to fits the abbreviation used there after as GFS

Authors: Thank you for this suggestion. We have modified the conclusion accordingly.

Reviewer 1. L 418, in the conclusion section  Plz add the dose and ratios of GFS for taken home massage.

Authors: Thank you for your comment. We have modified the sentence as follow:

“The inclusion of a combination of agro-industrial wastes: grapeseed, flaxseed and sea blackthorn meals in a ratio of 3:4:1 in the diet of weaned piglets have increased the content of omega-3 polyunsaturated fatty acids.”

Reviewer 1. In the conclusion section, don’t repeat the in conclusion, the title is adequate. 

Authors: Thank you for this suggestion. We have deleted the word conclusion

Reviewer 3 Report

Manuscript is well prepared and brings high degree of originality and novelty.

Authors may provide primers.

Authors may put coma after grape seed, line 72

Authors, may delete in conclusion, line 416

Author Response

The Authors wish to thanks the Reviewer for the appreciations concerning the quality of the manuscript and for the comments and the suggestion that will improve considerably the final form of the manuscript.

Reviewer 2. Authors may provide primers.

Authors: Thank you for your comment. We have already published the sequences of primers in our previous papers as already stated in the M&M. We consider that the duplication of the information will not bring a plus value of this paper as these information can be easily found in our previous paper and will charge without any utility the information already dense of the manuscript.

Reviewer 2. Authors may put coma after grape seed, line 72

Authors: The suggested correction was done.

Reviewer 2. Authors, may delete in conclusion, line 416

Authors: Thank you for this suggestion. We have deleted the word conclusion
